# GRACE: GRadient-based Active Learning with Curriculum Enhancement for Multimodal Sentiment Analysis

Xinyu Li*
MoE Key Laboratory of Brain-inspired Intelligent
Perception and Cognition, University of Science and
Technology of China
Hefei, China
xinyli@mail.ustc.edu.cn

Wenqing Ye*
University of Science and Technology of China
Hefei, China
wenqy@mail.ustc.edu.cn

Yueyi Zhang
MoE Key Laboratory of Brain-inspired Intelligent
Perception and Cognition, University of Science and
Technology of China
Hefei, China
zhyuey@ustc.edu.cn

Xiaoyan Sun†
MoE Key Laboratory of Brain-inspired Intelligent
Perception and Cognition, University of Science and
Technology of China
Hefei, China
sunxiaoyan@ustc.edu.cn

## Abstract

Multimodal sentiment analysis (MSA) aims to predict sentiment from text, audio, and visual data of videos. Existing works focus on designing fusion strategies or decoupling mechanisms, which suffer from low data utilization and a heavy reliance on large amounts of labeled data. However, acquiring large-scale annotations for multimodal sentiment analysis is extremely labor-intensive and costly. To address this challenge, we propose **GRACE**, a **GR**adient-based **A**ctive learning method with **C**urriculum **E**nhancement, designed for MSA under a multi-task learning framework. Our approach achieves annotation reduction by strategically selecting valuable samples from the unlabeled data pool while maintaining high-performance levels. Specifically, we introduce *informativeness* and *representativeness* criteria, calculated from gradient magnitudes and sample distances, to quantify the active value of unlabeled samples. Additionally, an *easiness* criterion is incorporated to avoid outliers, considering the relationship between modality consistency and sample difficulty. During the learning process, we dynamically balance sample difficulty and active value, guided by the curriculum learning principle. This strategy prioritizes easier, modality-aligned samples for stable initial training, then gradually increases the difficulty by incorporating more challenging samples with modality conflicts. Extensive experiments demonstrate the effectiveness of our approach on both multimodal sentiment regression and classification benchmarks.

*Both authors contributed equally to this research.
†Corresponding author.

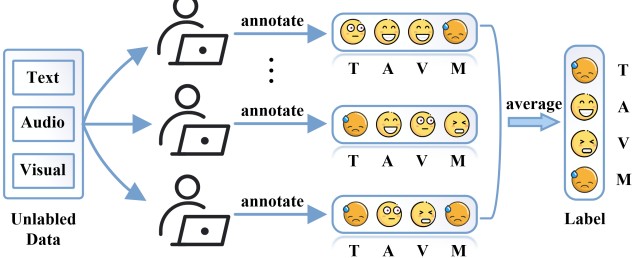

**Figure 1: An illustration of the labor-intensive annotation process for multimodal multi-task sentiment analysis, requiring multiple annotators to label each sample across text (T), audio (A), visual(V), and multimodal (M), highlighting the need for active learning to reduce annotation costs.**

## CCS Concepts

• **Information systems** → **Sentiment analysis**; *Multimedia information systems*.

## Keywords

Multimodal Sentiment Analysis, Active Learning, Curriculum Learning

**ACM Reference Format:**
Xinyu Li, Wenqing Ye, Yueyi Zhang, and Xiaoyan Sun. 2024. GRACE: GRadient-based Active Learning with Curriculum Enhancement for Multimodal Sentiment Analysis. In *Proceedings of the 32nd ACM International Conference on Multimedia (MM '24), October 28-November 1, 2024, Melbourne, VIC, Australia.* ACM, New York, NY, USA, 10 pages. https://doi.org/10.1145/3664647.3681617

## 1 Introduction

Sentiment analysis is a widely studied area that infers individuals' emotional states or tendencies by analyzing data. However, relying solely on a single modality for sentiment analysis presents inherent limitations [23]. The derived sentiment is often constrained and susceptible to signal noise, leading to one-sided, ambiguous, or even

contradictory outcomes. In contrast, multimodal sentiment analysis (MSA) emerges as a research field that leverages data from multiple modalities, such as text, audio, and visual sources, to analyze emotions more comprehensively. By integrating information from diverse modalities, MSA closely resembles real-world scenarios, resulting in a more accurate representation of expressed emotions.

In recent years, considerable efforts have been dedicated to developing fusion strategies and disentangling modality features to enhance sentiment analysis precision. Various fusion techniques have been explored, including early fusion [39], late fusion [50], and hybrid fusion [6], aiming to leverage the complementary strengths of each modality. In addition to fusion strategies, there has been a focus on disentangling modality features to uncover the distinct contributions of each modality in MSA [10, 17, 41].

While the potential of these methods in MSA is clear, their data inefficiency limits performance based on available dataset size. Despite extensive efforts to establish new MSA benchmarks [25], the number of labeled samples remains insufficient compared to other multimodal tasks. Although acquiring a large number of video samples is feasible in the multimedia era, labeling them poses a challenge, as shown in Figure 1. Given the subjectivity and ambiguity inherent in sentiment analysis, each sample in MSA often requires multiple annotators [46, 47] to derive a reliable average sentiment score, significantly increasing the overall cost of dataset annotation. Active learning has shown promise in various data-insufficient domains [14, 15, 19, 31] by prioritizing valuable unlabeled samples to reduce costs. However, most active learning methods struggle in multimodal tasks as they are designed for unimodal contexts, thus ignoring the complex relationships among modalities.

In this paper, we propose a **GR**adient-based **A**ctive learning method with **C**urriculum **E**nhancement (**GRACE**) that utilizes gradient embeddings in a multi-task learning framework to select multimodal samples beneficial for multimodal sentiment analysis. Our approach achieves data efficiency by annotating unlabeled samples deemed most valuable to the model. To determine the selection order of samples, we employ both an Active Value Estimator and an Easiness Estimator. These estimators consider the relationship between unimodal and multimodal gradient embeddings for each sample. Specifically, we assess *informativeness* by evaluating the magnitude of gradients, while *representativeness* is determined by accumulating the distances between gradients of different samples. Furthermore, to avoid selecting outliers, we incorporate an *easiness* criterion. It is hypothesized that samples demonstrating high consistency between unimodal and multimodal gradient embeddings are easier to learn. By adjusting the weight of these estimators, the model learns from easier to more challenging samples in a progressive order. Our contributions can be summarized as follows:

- We propose a novel gradient-based active learning method, GRACE, specifically for MSA under a multi-task learning framework. Our approach mitigates the difficulty of data annotation in MSA by selecting valuable unlabeled samples for labeling.
- We design three key criteria, informativeness, representativeness, and easiness, to assess the active value and sample difficulty for establishing the prioritization of sample selection. Additionally, the curriculum learning principle is incorporated to dynamically

adjust the evaluation approach across different active learning phases.
- We extensively evaluate the performances of our proposed method across various datasets for MSA. Comprehensive experiments demonstrate the superiority of our approach over existing active learning baselines, achieving higher performance with fewer labeled data.

## 2 Related Works

### 2.1 Multimodal Sentiment Analysis

Multimodal sentiment analysis involves the evaluation of text, audio, and visual data from a video clip to derive a quantitative representation of the expressed sentiment.

Most researchers have focused on the delicate design of networks to fuse unimodal data. Zadeh *et al.* [45] achieved tensor fusion through a 3-fold Cartesian product on unimodal embeddings, marking a significant advancement in multimodal fusion research of MSA. Following the introduction of the self-attention mechanism, cross-modal attention for fusion and modality adaptation has gained popularity [21, 32, 38, 43]. Decoupling modality information has also become a mainstream approach. MISA [10] proposes a shared encoder along with three specialized encoders, mapping modalities to modality-invariant and modality-specific subspaces. Subsequently, numerous studies have emerged integrating the decoupling of modality features with techniques such as distillation learning and contrastive learning [17, 41, 42].

Yu *et al.* [44] presented the SIMS dataset for sentiment analysis in Chinese, including annotations for both unimodal and multimodal aspects, and developed a multimodal multi-task learning framework that leverages unimodal labels. Liu *et al.* [20] relabeled and expanded the SIMS dataset to augment nonverbal signals in MSA. The CHERMA dataset is introduced by Sun *et al.*, which provides labels for unimodal and multimodal [34]. Their proposed layer-wise multimodal transformer model is also a multi-task learning framework. Despite the demonstrated effectiveness of multi-task learning frameworks in various experiments, it is crucial to tackle the issue of the increased labeling burden caused by this framework.

### 2.2 Active Learning

Pool-based Active Learning aims to maximize performance and minimize labeling costs by selecting informative samples from an unlabeled dataset. However, focusing solely on informativeness is insufficient due to data redundancy and outliers.

Informative-based active learning approaches assess the information content inherent in given samples in various ways. Classical machine learning methods like maximum entropy calculation [27] and Expected Model Change Maximization (EMCM) [3] remain widely used. BALD [7] and subsequent optimized methods [16, 37] utilize Bayesian neural networks to estimate uncertainty approximately. Additionally, Gao *et al.* [8] augmented unlabeled data and assessed sample informativeness based on the loss of consistency between augmented samples. Representative-based active learning methods tackle data redundancy from a distributional perspective, focusing on the similarity between unlabeled and labeled samples using different distance metrics [22, 29]. Furthermore, Sinha *et al.*

introduced the VAE model [33] for active learning, prioritizing unlabeled samples with low correlation to labeled ones.

Recent studies have explored explicit or implicit strategies to balance informative-based and representative-based sample selection. BoostMIS [48] selects a union of unstable and uncertain samples, while DBAL [49] identifies highly uncertain samples and then chooses representative ones. BADGE [1] maps samples into the gradient embedding space, leveraging gradient magnitude and the k-means++ algorithm for diversity and uncertainty. Furthermore, Shen *et al.* [30] extended BADGE with their BMMAL for modality balance in multimodal classification tasks. Nevertheless, current methods often struggle to simultaneously address data redundancy and outliers, limiting their effectiveness in multimodal tasks.

### 2.3 Curriculum Learning

Curriculum learning (CL) and active learning (AL) both emphasize sample selection during training. While AL seeks optimal results with fewer samples, CL aims to enhance performance and accelerate convergence through effective data selection strategies.

Curriculum learning (CL), first introduced by Bengio *et al.* [2], is inspired by human education that learns from easy to hard. CL has been widely applied in various fields, including natural language processing (NLP) [26, 36], computer vision (CV) [9, 12] , and reinforcement learning (RL) [5, 28]. For example, in neural machine translation, CL has reduced training time by 70% and improved performance by 2.2 BLEU points compared to traditional methods [26]. Similarly, Jiang *et al.* [13] reported a 45.8% relative improvement in MAP and faster convergence in multimedia event detection with CL. In reinforcement learning, CL helps agents tackle complex goal-directed problems that would otherwise be unsolvable [5].

Recently, several works have emerged that combine AL and CL. Lin *et al.* [18] used self-paced learning and active learning independently, while Tang *et al.* [35] proposed a batch mode AL approach that considers both the potential value and easiness. However, both methods have limitations in deep learning tasks, focusing mainly on machine learning perspectives. In histological tissue classification [11] and remote sensing image classification [24], researchers have integrated AL and CL to tackle issues like imbalanced performance and insufficient data. Despite the widespread use of CL and AL, their applicability in multimodal settings remains unexplored.

## 3 Methodology

In this section, we introduce the details of the proposed GRACE (GRadient-based Active learning with Curriculum Enhancement), a novel gradient-based active learning method designed for MSA under a multi-task learning framework. GRACE consists of an Active Value Estimator and an Easiness Estimator, assessing sample value according to three criteria: informativeness, representativeness, and easiness, which jointly guide the model towards selecting high-quality data for annotation and training. The overall structure of GRACE is illustrated in Figure 3.

### 3.1 Problem Definition

The objective of MSA is to obtain the sentiment labels $y_m$ from three modality representations $\{x_t \in \mathbb{R}^{l_t \times d_t}, x_a \in \mathbb{R}^{l_a \times d_a}, x_v \in \mathbb{R}^{l_v \times d_v}\}$, representing text, audio, and visual, respectively. Here,

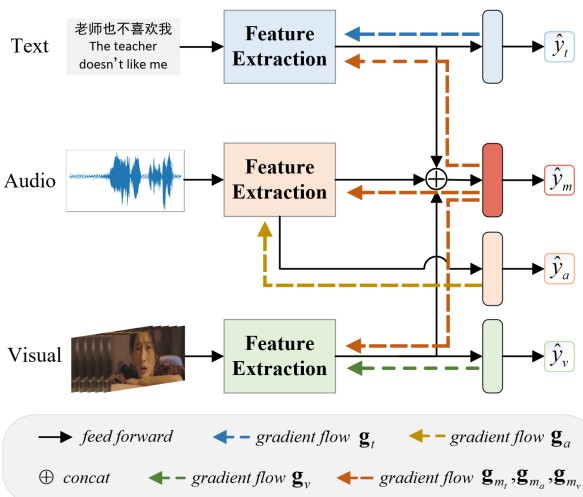

**Figure 2: An illustration of the multimodal multi-task learning framework. The dashed line with arrows indicates the direction of the gradient backpropagation for each task.**

$l_k$ and $d_k$ denote the sequence length and feature dimension for modality $k$, respectively, where $k \in K = \{t, a, v\}$. We also define $K^* = \{t, a, v, m\}$ for later use. Within the framework of multi-task learning, each sample has three unimodal sentiment labels ($y_t$, $y_a$, $y_v$), resulting in four sentiment labels per sample. During our active learning procedure, we commence with a pre-defined labeled set $L = \{(x_t^i, y_t^i); (x_a^i, y_a^i); (x_v^i, y_v^i); y_m^i\}_{i=1}^{N}$, which contains $N$ labeled samples. This set is utilized for the initial training of the task model, denoted as $F(\cdot; \theta)$, where $\theta$ represents the learnable parameters of the model. Subsequently, unlabeled samples are chosen from the unlabeled set $U$ using a query strategy. The AL annotator, Oracle then acquires their labels to construct an expanded labeled training set $L$. Concurrently, the queried unlabeled samples, denoted as $Q$, are removed from $U$. The task model is subsequently retrained on the updated labeled set $L$. This iterative procedure continues until the annotation budget is exhausted or the specified termination conditions are reached.

### 3.2 Multimodal Multi-task Learning Framework

Our GRACE is designed for MSA under a multi-task learning framework. We adopt the same architecture for multimodal multi-task learning [20], i.e., a late-fusion approach, as depicted in Figure 2. The network comprises three feature extraction networks: BERT for text, a linear layer followed by stacked bidirectional LSTM for audio, and a similar structure for visual. The extracted features $f_t \in \mathbb{R}^{h_t}, f_a \in \mathbb{R}^{h_a}, f_v \in \mathbb{R}^{h_v}$ from each modality are concatenated to obtain the fusion modality feature $f_m \in \mathbb{R}^{(h_t+h_a+h_v)}$ and processed through a multilayer perceptron (MLP) to yield the fused multimodal sentiment $\hat{y}_m$. Additionally, the features extracted from individual modality are passed through a fully connected (FC) layer to produce the corresponding unimodal sentiment $\{\hat{y}_t, \hat{y}_a, \hat{y}_v\}$. Given that our method does not rely on the specific output format of the network, it can be applied to various tasks. Specifically, for the regression task, we utilize the L1 loss function as supervision,

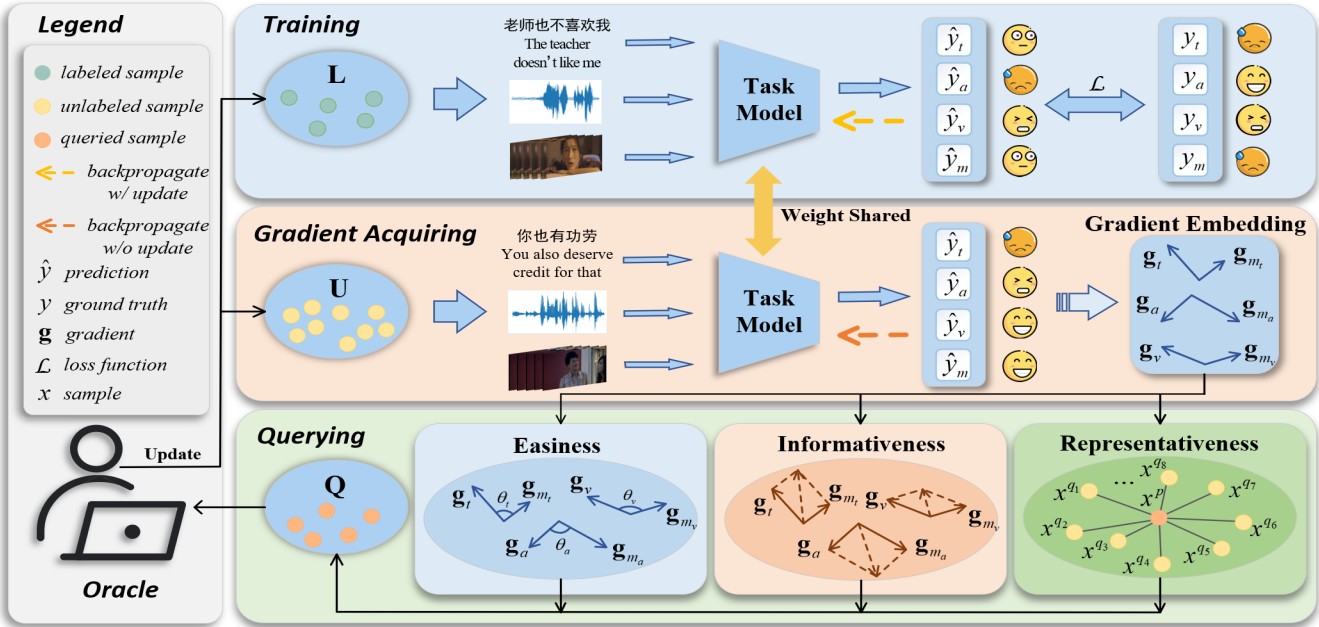

**Figure 3: An overview of the proposed GRACE framework. The letters L, U, and Q represent labeled pool, unlabeled pool, and queried samples, respectively. (1) In the training phase, the model is trained on available labeled data. (2) In the gradient acquiring phase, gradients are computed for unlabeled samples. (3) In the querying phase, unlabeled samples are queried based on three active learning criteria: easiness, informativeness, and representativeness. Selected samples are annotated by the Oracle, added to the labeled pool (L), and removed from the unlabeled pool (U). This completes one iteration of the AL cycle.**

calculated as follows:

$$\mathcal{L}_k = \frac{1}{N} \sum_{i=1}^{N} |\hat{y}_k^i - y_k^i|, k \in K^* \qquad (1)$$

where $\hat{y}_k^i$ and $y_k^i$ denote the predicted output and the ground truth for the $i$-th sample, respectively. $N$ is the size of the labeled pool. For the classification task, we employ the cross-entropy loss, defined as:

$$\mathcal{L}_k = -\frac{1}{N} \sum_{i=1}^{N} y_k^i \log \hat{y}_k^i, k \in K^* \qquad (2)$$

where $\hat{y}_k^i$ is the probability prediction generated by the MLP output of the model after applying a softmax function. Therefore, the final optimization objective of the model is formulated as the weighted sum of the losses associated with both multimodal and unimodal tasks:

$$\mathcal{L} = \sum_k w_k \mathcal{L}_k, k \in K^* \qquad (3)$$

where $w_k$ is the weight of $L_k$ for modality $k$. Specifically, $w_k$ are 0.2, 0.8, 0.4 and 1.0 for $k = t, a, v, m$, respectively.

To facilitate the subsequent formulation of expressions in a multimodal multi-task learning framework, we provide the process of gradient acquisition of the corresponding network layers for further analysis. Our approach focuses on the flattened gradient embeddings of the last layer in each unimodal feature extraction network, considering both unimodal and multimodal tasks. The gradient $\mathbf{g} = \frac{\partial \mathcal{L}}{\partial W_k}$ is obtained by forwarding unlabeled sample $x$

through the current model, computing the pseudo label $\tilde{y}(x)$ and backpropagating the loss on $(x, \tilde{y}(x))$, where $W_k$ is the weight of the last layer for the network of modality $k$. We denote the gradients resulting from the unimodal task loss for text, audio, and visual feature extraction networks as $\{\mathbf{g}_t, \mathbf{g}_a, \mathbf{g}_v\}$, while $\{\mathbf{g}_{m_t}, \mathbf{g}_{m_a}, \mathbf{g}_{m_v}\}$ represents the gradients of the multimodal task loss for the corresponding feature extraction network. In the training phase, the actual gradient applied to the unimodal network parameters is calculated as the sum of the gradients generated by both unimodal and multimodal tasks.

## 3.3 Gradient-based Active Value Estimator

In active learning, estimating the value of data samples for the task model is crucial. Traditionally, this evaluation is conducted from the perspectives of informativeness and representativeness, often considering only unimodal data. These approaches neglect the complex interactions among different modalities and the critical role of fusion modality in MSA. Inspired by the gradient descent algorithm, we leverage the gradient embeddings from the task model to quantify both informativeness and representativeness within each sample.

Informativeness reflects the uncertainty of the current model's predictions for a given sample. To assess the information content of multimodal samples, we develop a gradient-based informativeness criterion. The gradient embeddings capture not only the current uncertainty of the model but also potential parameter update directions, thereby serving as a powerful tool for guiding the active

learning process. Specifically, we combine individual modality gradient embedding $\{\mathbf{g}_t, \mathbf{g}_a, \mathbf{g}_v\}$ with the fusion modality gradient $\{\mathbf{g}_{m_t}, \mathbf{g}_{m_a}, \mathbf{g}_{m_v}\}$ respectively, calculating the magnitude as the uncertainty for each sample. The informativeness score for unlabeled sample $x^p \in U$ is calculated as:

$$\hat{s}_i(x^p) = \sum_{k \in K} \|\mathbf{g}_k(x^p) + \mathbf{g}_{m_k}(x^p)\|_2 \qquad (4)$$

where $\|\cdot\|_2$ denotes the $L_2$ norm. A higher informativeness score suggests that the sample can provide more valuable information to the current model. For each unimodal gradient $\mathbf{g}_k(x^p)$, the corresponding fusion modality gradient $\mathbf{g}_{m_k}(x^p)$ is incorporated to represent the potential change in the unimodal network, highlighting the central role of the fusion modality. From this perspective, the model prioritizes unlabeled samples based on their informativeness scores, with the instance eliciting the maximum magnitude of gradient embeddings deemed most informative.

While the informativeness criterion facilitates the selection of samples with the highest information content, it does not account for the overall distribution of the unlabeled data. Specifically, it overlooks sample diversity, which may lead to the selection of highly informative yet redundant samples. To address this, we propose a gradient-based representativeness criterion.

Similar to the informativeness criterion, the representativeness criterion focuses on the fusion modality gradient, combined with each unimodal gradient. However, instead of emphasizing gradient magnitude, the distance between different sample gradient embeddings is considered. This criterion aims to query more diverse data by prioritizing samples with significant differences. The representativeness score for each sample is calculated as the sum of distances between its gradient and other unlabeled samples' gradients using Euclidean distance:

$$\hat{s}_r(x^p) = \sum_{k \in K} \sum_{x^q \in U \setminus x^p} Dist(\mathbf{g}_k(x^p) + \mathbf{g}_{m_k}(x^p), \mathbf{g}_k(x^q) + \mathbf{g}_{m_k}(x^q))$$
$$(5)$$

where $Dist(\cdot, \cdot)$ denotes Euclidean distance, and $x^q$ denotes the sample except $x^p$ in the unlabeled dataset. A higher representativeness score suggests greater sample diversity and potential value to the model.

## 3.4 Easiness Estimator based on Gradient Harmonization

The Gradient-based Active Value Estimator effectively estimates the value of samples in active learning, taking into account both their informativeness and representativeness. It aims to identify the most beneficial samples for the current state of the model. Nevertheless, solely relying on Active Value Estimator tends to favor samples with modality inconsistencies such as sarcasm, which can introduce novel information but significant training difficulties [40].

These selected samples are typical problematic samples in active learning, namely outliers that deviate substantially from the underlying data distribution and lead to resource wastage. To address this potential issue and provide a more comprehensive estimation of sample value in the context of MSA, we propose an Easiness Estimator based on gradient harmonization. Specifically, the Easiness Estimator, defined by the following equation, quantifies the overall

gradient consistency across modalities for a given sample $x^p \in U$,

$$\hat{s}_e(x^p) = \sum_{k \in K} Sim(\mathbf{g}_k(x^p), \mathbf{g}_{m_k}(x^p)) \qquad (6)$$

where $Sim(\cdot, \cdot)$ represents the cosine similarity function. The Easiness Estimator calculates the cumulative cosine similarity between the gradients of the modality-specific tasks $\mathbf{g}_k(x^p)$ and their counterparts $\mathbf{g}_{m_k}(x^p)$ in the multimodal task. A higher easiness score indicates stronger alignment or harmony among the gradients, suggesting that the samples are more consistent and easier to learn. By contrast, a lower value implies modality conflicts that may hinder the learning process and should be deprioritized during sample selection. Since each unimodal gradient needs to be compared with the multimodal gradient, it can be observed that the calculation of easiness places more emphasis on the fusion modality, quantifying the harmonization of gradients for unlabeled samples.

To ensure comparability among the informativeness, representativeness, and easiness criteria, which may have varying score ranges, we employ min-max normalization. This technique rescales the scores to a unified range of $[0, 1]$ by $s_j = \frac{\hat{s}_j - \hat{s}_j^{min}}{\hat{s}_j^{max} - \hat{s}_j^{min}}$, where $\hat{s}_j^{min}$ and $\hat{s}_j^{max}$ respectively represent the minimum and maximum values of $\hat{s}_j$ for $j \in \{i, r, e\}$ across all samples.

## 3.5 Dynamic Curriculum-enhanced Active Strategy

The informativeness, representativeness, and easiness criteria presented above can estimate the sample value comprehensively from various dimensions. By considering these criteria, we can effectively mitigate common challenges in active learning, namely sample redundancy and outliers.

To enhance the Active Value Estimator and the Easiness Estimator by curriculum learning principle, we introduce a curriculum factor, denoted as $\alpha$. We combine these criteria to obtain the final score $s$ for each unlabeled sample using the following equation:

$$s(x^p) = s_e(x^p) \cdot \alpha + s_i(x^p) \cdot s_r(x^p) \qquad (7)$$

The curriculum factor $\alpha$ serves as a balancing mechanism between sample difficulty and the active value.

Moreover, the same level of sample difficulty may hold varying worth across different active learning phases. In the initial stages, when training data is limited and the learned decision boundaries may deviate from the actual boundaries, we assign a higher value to $\alpha$. This allows the active learning strategy to prioritize simpler samples, facilitating more efficient model learning. As training progresses and the network becomes more mature, we gradually decrease the value of $\alpha$ by curriculum decay $\alpha_d$, shifting the active learning process towards selecting more challenging samples. This progressive increase in sample difficulty aims to enable the model to achieve higher performance levels, mimicking the gradual learning curve exhibited by humans. The specific updating rule for $\alpha$ at the $t$-th query round is given by:

$$\alpha_t = \alpha_{init} - (t - 1) \cdot \alpha_d \qquad (8)$$

where $\alpha_{init}$ denotes the initial curriculum factor.

By integrating the curriculum learning principle into our active learning framework, we strike a delicate balance between sample

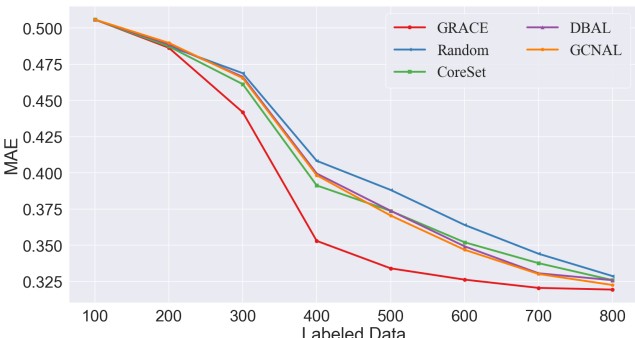

Figure 4: Model performance comparison of regression task on SIMSv2. The initial labeled pool contains 100 samples and increases to 800 samples by seven AL cycles. A smaller MAE score indicates better performance.

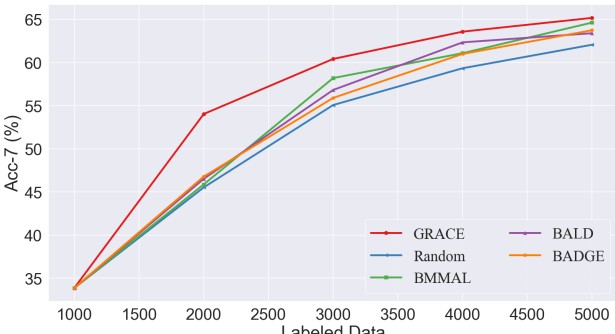

Figure 5: Model performance comparison of classification task on CHERMA. The initial labeled pool contains 1000 samples and increases to 5000 samples by four AL cycles. A larger Acc-7 score indicates better performance.

difficulty and active value. This dynamic selection strategy avoids the common disadvantages of active learning and offers a more effective training manner for MSA.

## 4 Experiments

### 4.1 Settings

We conduct experiments on two publicly available MSA datasets that provide unimodal annotations: SIMSv2 [20] and CHERMA [34]. The former is used for regression and the latter for classification. See details about datasets in Appendix A. Following previous works, we evaluate performance on the regression task using metrics: mean absolute error (MAE), binary accuracy (Acc-2), three-class accuracy (Acc-3), five-class accuracy (Acc-5), Pearson correlation (Corr), and F1 score. For classification, we employ a seven-class accuracy (Acc-7) metric. All experiments are conducted in PyTorch on an RTX 3090 GPU with 24GB memory. Hyper-parameters of the task model match those in [20]. The initial labeled pool is randomly sampled from the training set. To mitigate overfitting, we set the training epoch to 5, using the model from the last epoch as optimal without utilizing the labeled validation set. Our experiments on SIMSv2 and CHERMA datasets start with 100 and 1000 initial samples, respectively. The curriculum factor $\alpha$ and decay $\alpha_d$ are set to 1 and 0.1 respectively, with $\alpha$ updated at the end of each query round.

We compare our approach against several widely-used baseline methods in active learning. **CoreSet** [29] minimizes the distance between unlabeled samples and their nearest labeled counterparts after incorporating a set number of samples into the training set. **GCNAL** [4] uses a graph convolution network to distinguish labeled and unlabeled samples, selecting those that significantly differ from the labeled ones. **DBAL** [49] selects samples closest to the center point after clustering. **BALD** [7] estimates result uncertainty via Bayesian neural networks, maximizing mutual information between model outputs and parameters. **BADGE** [1] implicitly considers uncertainty and diversity with gradient embedding of samples. Furthermore, **BMMAL** [30] extends BADGE by achieving modality balance for multimodal classification tasks. Additionally, **Random** sampling is an unbiased method to choose samples from

the unlabeled pool. Since some methods are task-specific, we compare our approach with CoreSet, GCNAL, DBAL, and Random on the regression task, while conducting comparisons with BALD, BADGE, BMMAL, and Random on the classification task.

### 4.2 Results and Analysis

We present a detailed analysis comparing the performance of our method with other state-of-the-art active learning methods in both regression and classification tasks. To ensure fairness, all methods are evaluated with the same initial pool and task model, with results averaged over three trials using different random seeds.

**Regression Task on SIMSv2.** The performance comparison of the sentiment regression task on the SIMSv2 dataset is shown in Figure 4. As the number of labeled instances reaches 800, about 30% of the training set, our GRACE outperforms others. Remarkably, it achieves a performance comparable to Random sampling with just 500 samples, saving *1200 annotations (37.5%)* within the multi-task framework which requires four labels per sample. Compared with DBAL and CoreSet, our method also saves approximately *800 annotations (12.5%)*. This discrepancy arises because these methods are not designed to handle complex multimodal tasks efficiently. Furthermore, our method exhibits a more rapid initial decline in MAE, thanks to the easiness criterion that prioritizes samples with consistent modalities. This focus allows the model to quickly learn optimal initial parameters, mitigating the cold start problem associated with a small initial pool. Under curriculum-enhanced active learning, the model gradually incorporates more challenging samples, enhancing its generalization ability and robustness, and resulting in further performance improvements.

We also report multiple metrics for the compared methods in Table 1. The term "Full data" refers to the utilization of the complete training set, consisting of 2722 samples, to train the task model, while the other methods employed their own selected 800 samples for training. As evident from the table, our method outperforms other active learning methods across all metrics. Although GCNAL exhibits good performance in terms of MAE, its performance is inferior in metrics such as Acc-2 and Acc-3, indicating possible limitations in robustness. GRACE demonstrates differences of less

**Table 1: Experimental results of multiple metrics on SIMSv2. The results are reported when the number of selected samples reaches 800. The best results are highlighted in bold.**

| Methods | MAE ($\downarrow$) | Acc-2 ($\uparrow$) | Acc-3 ($\uparrow$) | Acc-5 ($\uparrow$) | F1 score ($\uparrow$) | Corr ($\uparrow$) |
|---|---|---|---|---|---|---|
| Full data | 0.295±0.005 | 82.11±0.68 | 75.08±0.68 | 52.51±1.21 | 82.17±0.69 | 74.60±0.63 |
| CoreSet | 0.326±0.007 | 80.11±1.79 | 69.99±1.22 | 47.71±0.97 | 80.20±1.77 | 69.76±1.39 |
| GCNAL | 0.322±0.012 | 79.85±1.26 | 70.89±2.05 | 49.52±2.31 | 79.89±1.31 | 69.32±2.01 |
| DBAL | 0.326±0.002 | 79.75±0.74 | 71.30±0.82 | 49.07±1.41 | 79.85±0.73 | 69.12±0.40 |
| Random | 0.329±0.010 | 79.98±1.49 | 70.25±1.07 | 48.00±1.23 | 80.03±1.48 | 69.15±1.53 |
| GRACE | **0.319±0.007** | **81.17±1.70** | **72.86±1.57** | **50.52±1.66** | **81.26±1.67** | **70.75±1.19** |

**Table 2: Ablation studies based on different variants of GRACE over SIMSv2. The best results are highlighted in bold.**

| Methods | MAE | Acc-2 | Acc-3 | Acc-5 | F1 score | Corr |
|---|---|---|---|---|---|---|
| GRACE-I | 0.346 | 78.88 | 67.50 | 45.74 | 78.50 | 69.22 |
| GRACE-IR | 0.323 | 79.95 | 71.47 | 48.58 | 80.05 | 70.00 |
| GRACE-IRE | **0.319** | **81.17** | **72.86** | **50.52** | **81.26** | **70.75** |

than **2.5%** across Acc-3, Acc-5, F1 score, and Corr metrics compared to the full data, especially with a difference of only **0.94%** observed in the Acc-2 metric. This highlights the ability of our method to achieve comparable performance using only a few labeled samples, approximately 30%.

**Classification Task on CHERMA.** Figure 5 presents the comparison results of the sentiment classification task. On the CHERMA dataset, our method achieves a **65.17%** seven-class accuracy with 5000 samples, about 30% of the training set, compared to **69.86%** with the full training set. As depicted in the figure, our method with 4,000 samples significantly outperforms Random, BALD, and BADGE, which saves **4,000 (20%) annotations** in a multi-task framework. Similar to the regression task, our method consistently outperforms other methods throughout the active learning phase, demonstrating its generalizability and task-agnostic nature. Compared to Random sampling, which has an accuracy of **62.05%**, our method shows a notable improvement, demonstrating greater potential when applied to large datasets. While BMMAL achieves 64.62% accuracy by balancing multiple modalities, it cannot surpass our method due to its tendency to select outliers, which we mitigate through carefully designed criteria.

## 5 Further Analysis

### 5.1 Ablation Studies

**Effectiveness of Criteria.** We conduct ablation experiments on the proposed three criteria, informativeness (I), representativeness (R), and easiness (E). The results of the ablation experiments shown in Table 2 illustrate the impact of different criteria on SIMSv2. GRACE-I and GRACE-IR are variants of GRACE in which sample scores are calculated using only informativeness or only informativeness and representativeness criteria. It can be observed that GRACE-IRE, the original GRACE, yields the highest performance on all evaluation metrics, with a 2.29% and 1.22% improvement in Acc-2 over the

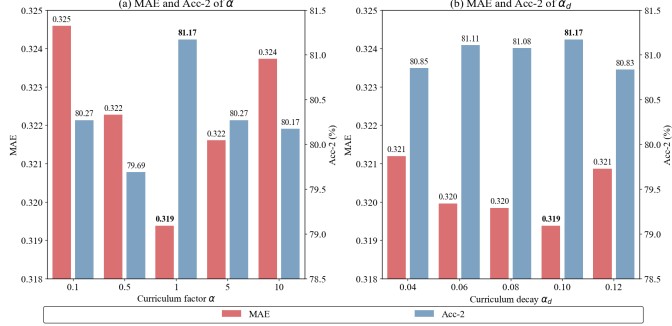

**Figure 6: Ablation studies with respect to different values of $\alpha$ and $\alpha_d$ on SIMSv2. The lowest MAE and highest Acc-2 values are highlighted in bold.**

other variants. It can also be noticed that the performance is even lower than Random sampling as shown in Table 1 when only the informativeness criterion is considered. This is because the informativeness criterion picks out a lot of redundant samples and prefers difficult samples with high uncertainty, which is not conducive to learning in the early stages of the model. The findings emphasize the importance of considering three criteria simultaneously.

**Effectiveness of Curriculum Enhancement.** To investigate the effect of different levels of curriculum learning enhancement on the model, we conduct experiments varying the values of $\alpha$ and $\alpha_d$ in Figure 6. Specifically, for the experiments concerning $\alpha$, we set $\alpha_d$ to be 0.1 times the value of $\alpha$. On the other hand, we fix $\alpha$ at a value of 1 when studying $\alpha_d$. From Figure 6(a), it is evident that increasing or decreasing the value of $\alpha$ relative to 1 leads to a decline in both the MAE and Acc-2 metrics. We hypothesize that this effect arises due to an excessive emphasis on samples with high modality consistency, which diminishes the role of representativeness and informativeness. Conversely, when the weight of the easiness criterion is too small, the influence of curriculum enhancement on the model declines, resulting in performance limitations. In Figure 6(b), we observe that the performances remain highly similar across multiple curriculum decay $\alpha_d$ values. This finding suggests that a slight change in the rate of sample difficulty would not significantly impact the overall curriculum learning progress. Consequently, we select the value of 0.1 for $\alpha_d$, considering it a reasonable choice without loss of generality.

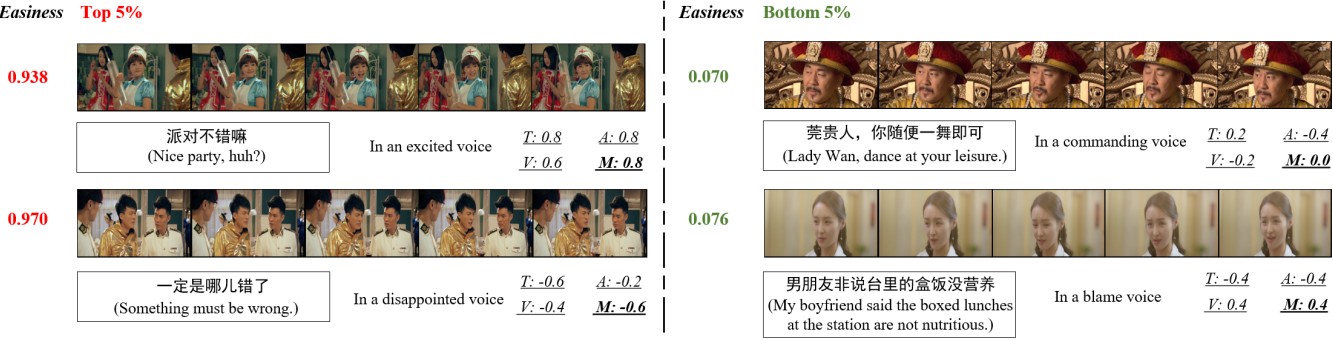

**Figure 7: Display of cases and easiness scores on SIMSv2. The top 5% sample shows modality consistency and reversely the bottom 5% samples show modality conflict. Below the visual images of each case are the text, the tone of the audio, and four sentiment labels.**

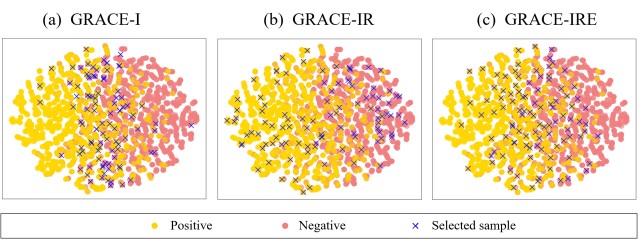

**Figure 8: Visualization of samples from different variants of GRACE on SIMSv2 dataset. The two kinds of dots represent positive and negative sentiment samples respectively, and the blue crosses represent selected samples at the 5th cycle.**

## 5.2 Case Analysis

We randomly select four samples from the top 5% and bottom 5% according to the easiness criterion, as shown in Figure 7. It is evident that the samples in the top 5% exhibit a high similarity between their unimodal and multimodal sentiments, with consistent positive or negative emotions. In contrast, the samples in the bottom 5% present significant modality conflicts, displaying lower consistency among modalities. Specifically, for the first sample on the left, the text modality expresses agreement with a positive sentiment, while the tone of the speaker conveys excitement, and the visual cues show a noticeable smile. Therefore, all four labels indicate high positive values, resulting in an easiness criterion score of 0.938. Conversely, the second sample on the right represents a typical example of mixed sentiments. Although the speaker appears visually smiling, the textual content and tone convey negative emotions. Such modality conflicts can significantly confuse the network, making it a challenging sample to process.

## 5.3 Data Distribution Visualization

To further analyze the distribution of selected samples for different variants, we visualize the selected samples by tSNE embeddings in Figure 8. As shown in Figure 8(a), it is evident that when considering only informativeness, the model prefers samples located near the decision boundary. These samples exhibit higher uncertainty, making them challenging to learn in the early stages of the network. Additionally, samples appear to overlap closely in some cases. These redundancies waste the annotation budget and provide a tiny performance boost to the task model. In Figure 8(b), when both informativeness and representativeness criteria are utilized together, the selected sample distribution becomes more uniform, with minimal redundancy. This observation supports our previous statement that combining the representativeness criterion leads to a more even distribution. Nevertheless, it is important to note that many samples far away from others, potentially outliers, are selected as candidates. Subsequently, Figure 8(c) shows that the introduction of the easiness criterion enables the model not only to achieve a more uniform distribution in the feature space but also to reduce the selection of potential outliers, thereby mitigating their impact on the model.

## 6 Conclusion

In this paper, we proposed GRACE, a novel gradient-based active learning method with curriculum enhancement, designed for multimodal multi-task sentiment analysis. To identify valuable samples for selection, we develop two estimators that assess active value and sample difficulty based on three criteria: informativeness, representativeness, and easiness. These criteria leverage the gradient embeddings of both unimodal and multimodal tasks, with a particular emphasis on the multimodal aspect. Enhanced by curriculum learning, GRACE dynamically adjusts the weight between active value and sample difficulty, ensuring that the task model learns progressively from easy to hard. The calculation of sample scores through these three criteria mitigates common issues in active learning such as sample redundancy and outliers. Comprehensive experiments on multimodal sentiment classification and regression tasks demonstrate the effectiveness of GRACE. However, there remains room for improvement in our approach, such as introducing an adaptive adjustment strategy for the curriculum factor or employing more sophisticated calculation methods for the criteria. Overall, our approach offers an effective solution to reduce annotation costs for MSA while maintaining model accuracy.

## Acknowledgments

This work was in part supported by the National Key R&D Program of China under grant 2020AAA0108600 and the National Natural Science Foundation of China (NSFC) under grants 62032006 and 62021001.

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
