# OpenReview forum: "GRACE: GRadient-based Active Learning with Curriculum Enhancement for Multimodal Sentiment Analysis"
_acmmm.org/ACMMM/2024/Conference — MM2024 Poster_

### Official Review · Reviewer_yD6y · 2024-05-14

**Rating:** 4
**Confidence:** 3

**Summary:**

In this work, the authors propose GRACE, a GRadient-based Active learning method with Curriculum Enhancement, designed for MSA under a multi-task learning framework. GRACE achieves annotation reduction by strategically selecting valuable samples from the unlabeled data pool while maintaining high-performance levels.

**Strengths:**

1.The authors describe the work clearly in this manuscript.
2.The theoretical analysis of different active learning strategies is sufficient.
3.The authors conducted extensive experiments and ablation experiments, and presented rich visual results.

**Limitations:**

1.In addition to comparing MAE and model accuracy, should the time complexity of active learning strategies be considered?
2.I sugguest the authors explain other gradient-based active learning methods in related work and analyze the differences from this work.

**Suitability:**

3

---

### Official Review · Reviewer_ywme · 2024-05-22

**Rating:** 4
**Confidence:** 3

**Summary:**

This paper proposes a new approach to MSA, which aims to solve the problem of low data utilization and reliance on large amounts of labeled data by introducing active learning and curriculum leaning. The entry point of the problem is better, and the experiment is more sufficient. I think the current version can be published.

**Strengths:**

Three indicators, Easiness, Informativeness and Representativeness, are proposed to measure the unlabeled samples, and the angle of each index is reasonable and can solve the proposed problems. At the same time, the relevant experiments have verified the effectiveness of each of the three indicators, and the visualization result is intuitive and clear.

**Limitations:**

Some of the details of the paper are not clear, such as the lack of explanation of Oracle and the relevant formulas for gradient calculation.
1. What is the difference between SIMSv2 and CHERMA datasets, and can you explain this?
2. Why are the baseline models employed on the two datasets not exactly the same?
3. Why are the experimental results on all the metrics shown only on the SIMSv2 dataset? How do these metrics look on CHERMA?
4. Regarding the curriculum factor, why does it tend to choose easy samples when set curriculum factor large? A higher easiness score suggesting that the samples are more consistent and easier to learn. Shouldn't you set the curriculum factor a small value at the beginning, so that the model are more inclined to choose easy samples with large easiness score?

**Suitability:**

3

---

### Official Review · Reviewer_UD16 · 2024-05-22

**Rating:** 2
**Confidence:** 2

**Summary:**

The paper introduces GRACE, a gradient-based active learning method with curriculum enhancement for multimodal sentiment analysis. GRACE selects samples based on three criteria, namely informativeness, representativeness, and easiness. This method has demonstrated effectiveness in extensive experiments on multimodal sentiment regression and classification benchmarks.

**Strengths:**

1. This paper is well-motivated. Broadening the criteria of sample selection in active learning is an interesting topic.

**Limitations:**

Major concerns:

The feasibility of the proposed method is my major concern. If the authors could provide more convincing information during the rebuttal, I would be glad to raise my score.

1. In lines 384-398 (left column), the authors mentioned that they calculate the gradient based on the pseudo label of the unlabeled data. However, as the pseudo label is also generated by this network, how could there be meaningful gradients for subsequent active value estimation?

2. Why did the authors choose to compute the distance between the gradients of two samples for the representativeness score, instead of directly computing the distance on their representations? Please provide more justifications for this and introduce experiments to compare the two solutions.

3. It would be much better if there could be a study on how to balance the three criteria during active learning.

Minor concerns:
N/A

**Suitability:**

3

---

### Official Review · Reviewer_obqr · 2024-05-24

**Rating:** 4
**Confidence:** 2

**Summary:**

The paper presents GRACE (GRadient-based Active Learning with Curriculum Enhancement), a novel method for multimodal sentiment analysis (MSA). The goal is to reduce annotation costs while maintaining high performance by strategically selecting valuable samples from an unlabeled data pool. GRACE operates within a multi-task learning framework and incorporates criteria such as informativeness, representativeness, and easiness to evaluate the value of unlabeled samples. The method dynamically balances sample difficulty and active value guided by the curriculum learning principle. Experiments on sentiment regression and classification benchmarks demonstrate GRACE's effectiveness in achieving high performance with fewer labeled samples compared to existing active learning methods.

**Strengths:**

1. The paper proposes a novel gradient-based active learning method that combines informativeness, representativeness, and easiness criteria. This comprehensive approach ensures a more efficient selection of valuable samples.
2. The study includes extensive experiments on multimodal sentiment regression and classification tasks, providing strong evidence of GRACE’s effectiveness compared to other state-of-the-art methods.

**Limitations:**

1. Why is the diversity of labeled samples not considered when calculating representativeness? Is it necessary to consider the diversity of samples in the pre-defined labeled set L?
2. I did not see any comparison of training efficiency between different methods in the experimental results. By observing Figures 4 and 5, GRACE and other baselines eventually converge to similar performance. Therefore, analyzing the overall training process complexity is very important to evaluate whether the current method has an efficiency advantage compared to other methods.
3. The paper lacks relevant detailed descriptions. For example, how is w_k in Equation 3 selected? Is it a hyperparameter? The experiment also lacks related analysis and discussion.

**Suitability:**

2

---

### Meta-Review · Area_Chair_o5T4 · 2024-07-01

**Recommendation:** Accept (Poster)
**Confidence:** 4

**Metareview:**

The authors introduce GRACE, a Gradient-based Active learning method with Curriculum Enhancement tailored for MSA within a multi-task learning framework. GRACE reduces annotation needs by strategically selecting valuable samples from the unlabeled data pool while maintaining high-performance levels. All the reviewers unanimously conclude the acceptance after rebuttal. However, there are still some minor issues that need to be solved. Please refer to the reviewers' comments and final thoughts below.